# Segment Anything for Histopathology

**Titus Griebel**[*1,2] (iD)                    TITUS.GRIEBEL@STUD.UNI-GOETTINGEN.DE
**Anwai Archit**[*2] (iD)                    ANWAI.ARCHIT@UNI-GOETTINGEN.DE
**Constantin Pape**[2,3,4] (iD)                    CONSTANTIN.PAPE@INFORMATIK.UNI-GOETTINGEN.DE

[1] *University Medicine Göttingen, Department of General, Visceral and Pediatric Surgery*

[2] *Georg-August-University Göttingen, Institute of Computer Science*

[3] *CAIMed - Lower Saxony Center for AI & Causal Methods in Medicine, Göttingen*

[4] *Cluster of Excellence Multiscale Bioimaging (MBExC), Georg-August-University Göttingen*

**Editors:** Accepted for publication at MIDL 2025

## Abstract

Nucleus segmentation is an important analysis task in digital pathology. However, methods for automatic segmentation often struggle with new data from a different distribution, requiring users to manually annotate nuclei and retrain data-specific models. Vision foundation models (VFMs), such as the Segment Anything Model (SAM), offer a more robust alternative for automatic and interactive segmentation. Despite their success in natural images, a foundation model for nucleus segmentation in histopathology is still missing. Initial efforts to adapt SAM have shown some success, but did not yet introduce a comprehensive model for diverse segmentation tasks. To close this gap, we introduce PathoSAM, a VFM for nucleus segmentation, based on training SAM on a diverse dataset. Our extensive experiments show that it is the new state-of-the-art model for automatic and interactive nucleus instance segmentation in histopathology. We also demonstrate how it can be adapted for other segmentation tasks, including semantic nucleus segmentation. For this task, we show that it yields results better than popular methods, while not yet beating the state-of-the-art, CellViT. Our models are open-source and compatible with popular tools for data annotation. We also provide scripts for whole-slide image segmentation.

**Keywords:** segment anything, histopathology, instance segmentation

## 1. Introduction

Nucleus segmentation is an important task in computational pathology, with applications in cancer grading, tumor micro-environment analysis and survival prediction (van der Laak et al., 2021). Most approaches rely on supervised learning, typically based on UNets (Ronneberger et al., 2015). Several methods, e.g. StarDist (Weigert and Schmidt, 2022) or HoVerNet (Graham et al., 2019), address domain-specific challenges, such as overlapping nuclei or joint segmentation and classification. HoVer-UNet (Tommasino et al., 2024), HoVerNeXt (Baumann et al., 2024), and InstanSeg (Goldsborough et al., 2024) focus on efficiency without sacrificing segmentation accuracy. Despite these developments, existing models struggle to generalize to new conditions, for example due to different staining protocols, imaging devices, or sample extraction quality. Adapting them to new data involves

---

* Contributed Equally

training new models, which requires domain expertise for precise and exhaustive annotations (drawing each nucleus in new training patches). While tools like QuPath (Bankhead et al., 2017) are popular for this task, the procedure remains tedious and resource intensive.

Recently, vision foundation models (VFM) for segmentation tasks have been introduced (Kirillov et al., 2023; Zou et al., 2024). They generalize to different imaging domains and support different segmentation tasks, including automatic and *interactive* segmentation. In particular, the Segment Anything Model (SAM) (Kirillov et al., 2023) enables interactive segmentation from user input. SAM has been successfully applied and adapted to biomedical imaging domains (Archit et al., 2025a; Cheng et al., 2023; Yue et al., 2024; Archit et al., 2025b), but so far has not been comprehensively adapted to histopathology, where the performance of the original SAM is limited (Deng et al., 2023; Zhang et al., 2023; Archit et al., 2025a; Xu et al., 2024). Prior work used SAM for specific tasks in histopathology. For example, CellViT (Hörst et al., 2024) used its pretrained encoder for semantic instance segmentation. However, it only addresses automatic segmentation, neglecting interactive segmentation, and the available models are trained on a single dataset, PanNuke (Gamper et al., 2019). Other approaches, such as (Xu et al., 2024) use SAM for semi-supervised instance segmentation. However, this approach does not leverage the full potential of interactive segmentation with SAM, which enables human-in-the-loop correction and data-specific finetuning (Archit et al., 2025a). More recently, BioMedParse (Zhao et al., 2024), which is based on SEEM (Zou et al., 2024), has demonstrated text-prompt based semantic segmentation, but lacks support for instance segmentation. In summary, a VFM for automatic and interactive nucleus segmentation in histopathology is missing.

Here, we present *PathoSAM*, a VFM for nucleus segmentation in histopathology. It supports interactive and automatic segmentation, based on finetuning SAM on a large and diverse annotated dataset. PathoSAM outperforms other SAM variants for interactive segmentation, is the new state-of-the-art for nucleus instance segmentation, and supports finetuning for semantic segmentation. PathoSAM is compatible with the user-friendly tools QuPath (Bankhead et al., 2017) and $\mu$SAM (Archit et al., 2025a). We provide scripts to automatically segment whole-slide-images, together with the rest of our code, at https://github.com/computational-cell-analytics/patho-sam. Our contributions and a summary of results are shown in Fig. 1.

## 2. Methods

We review interactive (Sec. 2.1) and automatic (Sec. 2.2) instance segmentation with SAM and $\mu$SAM, which form the basis of PathoSAM, our VFM for histopathology (Sec. 2.3). We also study automatic semantic segmentation based on PathoSAM (Sec. 2.4). Our models are trained and evaluated on a large aggregated histopathology dataset (Sec. 2.5).

### 2.1. Segment Anything Model and Interactive Segmentation

SAM (Kirillov et al., 2023) has introduced a new formulation for interactive segmentation, where a user can provide input prompts, points (positive or negative), a bounding box or a low-resolution mask, to identify an object. The model then predicts the corresponding mask by processing the image with the image encoder, a vision transformer (ViT, (Dosovitskiy et al., 2021)), the prompts with the prompt encoder, and the outputs of image encoder

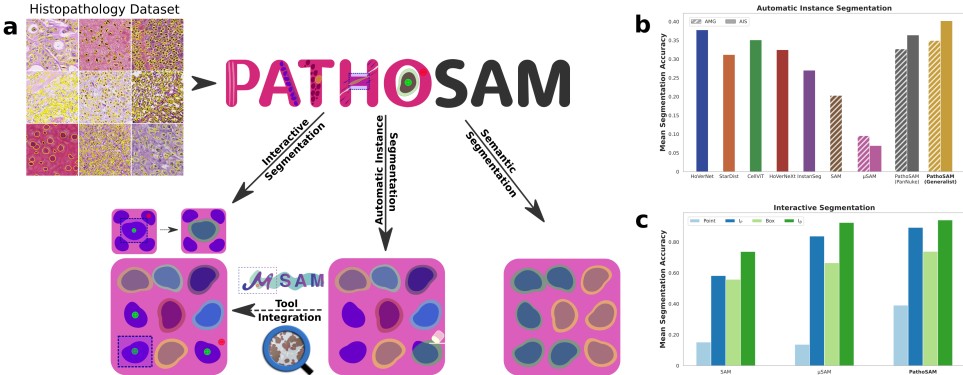

Figure 1: a) Overview of the PathoSAM: we train generalist models for nucleus segmentation on a combination of 6 datasets. The models support interactive and automatic instance segmentation, are integrated with user-friendly tools, and can be further finetuned for semantic segmentation. b) Automatic instance segmentation results of 12 datasets. PathoSAM (ViT-L) is the best model, despite evaluations favoring other models by choosing the best model per dataset (if multiple versions are available). c) Interactive instance segmentation results of 12 datasets. PathoSAM is the best model, especially for single point prompts.

and prompt encoder with the mask decoder. The predictions can be corrected by providing more prompts. See Fig. 5 for an overview of SAM's architecture.

SAM is trained on a large natural image dataset with annotations, using an objective that simulates interactive segmentation. In each iteration, this objective first samples prompts from an annotated mask, predicts the object mask, and then iteratively corrects the prediction with prompts sampled from the annotation. Predictions and annotations are compared with a loss function for each iteration, and the average loss is used for parameter updates. We use the training implementation of $\mu$SAM (Archit et al., 2025a).

To evaluate interactive segmentation, we automatically derive prompts from annotations to segment the object and then iteratively correct the segmentation, similar to the training objective. We perform 7 correction iterations. We compute the mean segmentation accuracy (App. A) between annotations and predictions for the initial prompt and corrections. We report the results for an initial point prompt, an initial box prompt, and the last correction iteration, when starting from a point ($I_P$), and when starting from a box ($I_B$).

## 2.2. Automatic Segmentation

SAM supports automatic instance segmentation by placing a grid of point prompts across the image, applying the model to each prompt to obtain segmentation masks, and removing overlapping or unlikely masks. This approach is called automatic mask generation (AMG).

$\mu$SAM (Archit et al., 2025a) introduces an alternative method for automatic instance segmentation, by adding a decoder, similar to UNETR (Hatamizadeh et al., 2022), to the model. It predicts foreground probabilities, the distance to the closest object center and the

distance to the closest object boundary. During training, the decoder is jointly optimized with the rest of the model, using targets derived from ground-truth annotations. During inference, the decoder outputs are processed with a seeded watershed to obtain an instance segmentation. $\mu$SAM has termed this approach automatic instance segmentation (AIS), and has shown that it improves over AMG in terms of segmentation quality and computational efficiency for microscopy data. We evaluate interactive segmentation (for SAM or variants) with AMG and, if available, AIS, using the mean segmentation accuracy, (App. A).

### 2.3. A vision foundation model for histopathology

In order to train a VFM model for histopathology, we adopt the training procedure of $\mu$SAM for joint training of interactive (Sec. 2.1) and automatic (Sec. 2.2) segmentation. We train different models based on the datasets for nucleus segmentation (Sec. 2.5): three generalist models trained on 6 datasets, one for each of the three image encoders (ViT-B, ViT-L and ViT-H), which we call PathoSAM generalists, and a ViT-B model trained on PanNuke. The training settings can be found in App. C.

### 2.4. Semantic Segmentation

We extend PathoSAM to semantic segmentation, to support the analysis of different cell types. We add a separate decoder, following the same design as in Sec. 2.2. See also Fig. 5 for the combined architecture. This decoder predicts one output channel per semantic class, including background, and is optimized based on the cross entropy loss. We train and evaluate it on PanNuke ((Gamper et al., 2019), Sec. 2.5), which provides semantic annotations for 5 classes. Here, we compare 4 different training strategies, which start from a PathoSAM generalist: (i) freeze the image encoder, initialize the new decoder from scratch; (ii) finetune the image encoder initialize the new decoder from scratch; (iii) freeze the image encoder, initialize the new decoder with the AIS decoder weights; (iv) finetune the image encoder, initialize the new decoder with the AIS decoder weights. Semantic segmentation is evaluated with the class-frequency weighted dice score, see App. A.

### 2.5. Data

We assemble 14 publicly available histopathology datasets to train and evaluate PathoSAM. We use 6 datasets (Vu et al., 2019; Graham et al., 2021; Kumar et al., 2017; Gamper et al., 2019; Schuiveling et al., 2025; Naylor et al., 2019) with tissue images stained with hematoxylin and eosin (H&E) staining and nucleus instance annotations to train our generalists. For each of these datasets we reserve a separate test split. An additional model is trained on PanNuke (Gamper et al., 2019), which is the most popular dataset for nucleus segmentation. This dataset is also used for semantic segmentation, since it provides semantic annotations for 5 classes. We use the remaining 8 datasets (Graham et al., 2019; Mahbod et al., 2021; Sirinukunwattana et al., 2017; Naji et al., 2024; Verma et al., 2021; Alemi Koohbanani et al., 2020; Mahbod et al., 2024; Wang et al., 2024) for out-of-domain evaluation, i.e to test our models on data which was not directly represented in the training set. Among these, two datasets contain different segmentation tasks, lymphocyte segmentation (Wang et al., 2024) and gland segmentation (Sirinukunwattana et al., 2017). They are used for separate evaluation experiments. Details on the datasets can be found in App. B.

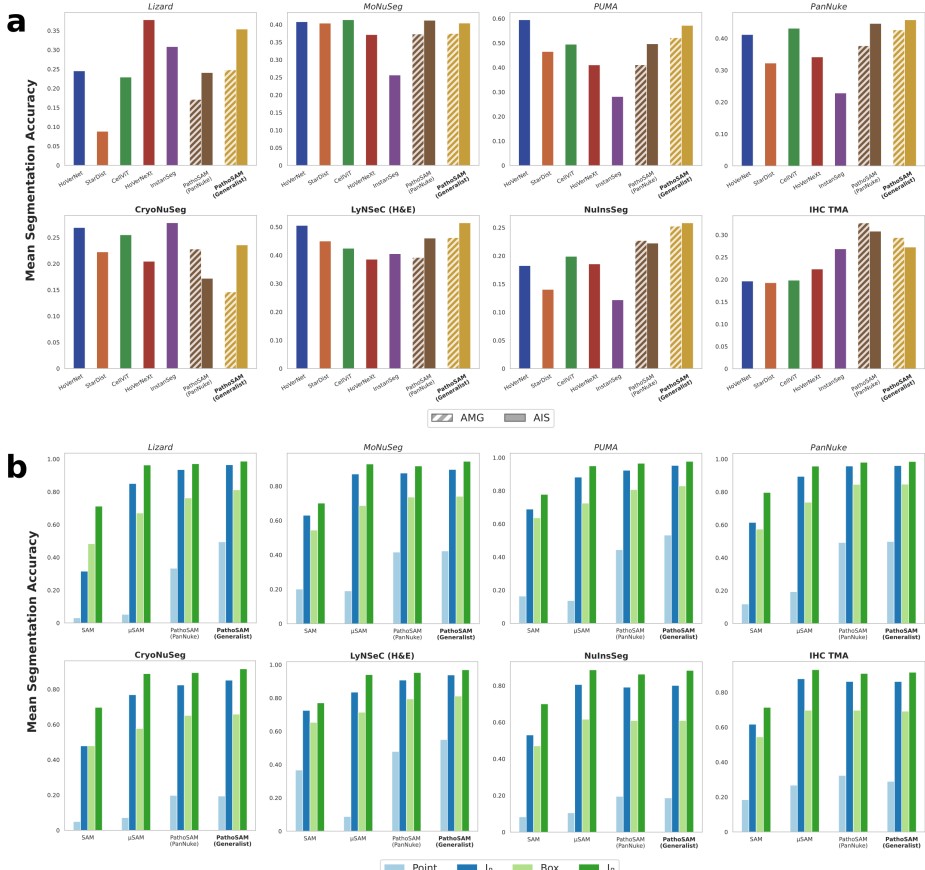

Figure 2: a) Automatic instance segmentation results for 8 datasets. Italic font indicates that the corresponding training split of the dataset was used for our generalist training, while bold font indicates that it was not. b) Interactive segmentation results for 8 datasets (same as in a)) for PathoSAM and other SAM variants. We report the interactive segmentation quality for initial point and box prompts as well as correction for 7 iterations after an initial point ($I_P$) and box ($I_B$).

## 3. Results

We compare PathoSAM to state-of-the-art methods for automatic instance segmentation (Sec. 3.1), interactive segmentation (Sec. 3.2), and for semantic segmentation (Sec. 3.3).

### 3.1. Automatic Instance Segmentation

We compare PathoSAM (generalist) with 5 different methods for automatic instance segmentation, HoVerNet (Graham et al., 2019), StarDist (Weigert and Schmidt, 2022), CellViT (Hörst et al., 2024), HoVerNeXt (Baumann et al., 2024), InstanSeg (Goldsborough et al., 2024), see App. D.1 details on the other methods. We evaluate these methods on 12 dif-

ferent datasets, 6 that were part of the training set for our generalists, using separate test splits for evaluation, and 6 that were not. The methods HoverNet, HoverNext and CellViT offer different model versions. For them, we evaluate all available versions and report the results for the best version per dataset. Fig. 2 a) shows the results on 8 selected datasets, for the ViT-L PathoSAM generalist. Fig. 1 b) shows the average over all datasets. The detailed results are given in App. D.1.

We observe that PathoSAM with AIS is the best overall model, *despite* our reporting favoring methods with multiple versions. On the "in-domain" datasets, PathoSAM is either the best or on par with the best method. For the "out-of-domain" datasets it shows remarkable generalization, outperforming other methods for NuInsSeg (H&E-stained images) and IHC-TMA (IHC-stained tissue microarrays). CryoNuSeg (cryo-sectioned H&E-stained images) is the only dataset where it underperforms, see Fig. D.1 for details. We include the comparison to PathoSAM (PanNuke), to investigate the advantage of training on multiple datasets. The effectiveness of this approach is validated, as the generalist model is on par or better for all datasets except IHC-TMA. We also see that AIS is superior to AMG in most cases. Hence this approach is clearly preferable, also due to its better efficiency (Archit et al., 2025a). The results in D.1 show that differences between ViT-B, ViT-L and ViT-H are small, with ViT-L reaching the highest score. Furthermore, we investigate the segmentation quality across overlap thresholds in Fig. 9 and the influence of post-processing parameters on the precision-recall trade-off in Fig. 10. We also study the segmentation results for neutrophils, which have complex nuclear shapes, in Fig. 11.

## 3.2. Interactive Instance Segmentation

We compare PathoSAM (generalist, ViT-B), with other SAM variants for interactive segmentation. Specifically, we compare it with the original SAM (Kirillov et al., 2023) and with $\mu$SAM (Archit et al., 2025a), which was finetuned on a large microscopy dataset. We use the same 12 datasets as before. The results for 8 selected datasets are reported in Fig. 2 b) and the overall results in Fig. 1 c). Detailed results are given in App. D.2. We use the evaluation procedure introduced by $\mu$SAM, see also Sec. 2.1, reporting the mean segmentation accuracy for segmentation with a point prompt, a box prompt, iterative correction after an initial point, and iterative correction after an initial box.

PathoSAM outperforms the other models when using a single point or box prompt. The segmentation quality after iterative correction of PathoSAM and $\mu$SAM are on par; they are clearly better than SAM. Notably, their segmentation quality after correction is almost perfect. The strong performance of $\mu$SAM demonstrates that SAM variants can generalize beyond their training set for interactive segmentation. This is also apparent for CryoNuSeg, where PathoSAM improves drastically through iterative correction.

## 3.3. Other Segmentation Tasks

We also evaluate PathoSAM on three other segmentation tasks. First, we evaluate segmentation of different structures, lymphocytes in IHC-stained tissue images (Alemi Koohbanani et al., 2020) and glands in H&E-stained colon tissue images (Sirinukunwattana et al., 2017). Here, we compare the automatic and interactive segmentation with SAM, $\mu$SAM, and PathoSAM. We also finetune a new model, PathoSAM (specialist), on the training splits of

the respective datasets. Quantitative and qualitative results are shown in Fig. 3 a) and b). For lymphocytes, SAM, $\mu$SAM and PathoSAM (generalist) cannot automatically segment the data well, but provide good interactive segmentation. Conversely, the specialist model yields better automatic segmentation, but is worse for interactive segmentation. The reason for the difference in automatic segmentation is clear: the lymphocytes only make up a fraction of the nuclei in the images. In automatic segmentation, the non-specific models segment all nuclei, resulting in a bad score. The specialist model distinguishes lymphocytes from other nuclei, thus providing a better result. It is unclear why it performs worse in interactive segmentation. Overall, this task may be better formulated as a semantic instance segmentation problem, see also next paragraph, to avoid penalizing the segmentation of all nuclei. For gland segmentation, the specialist outperforms all other SAM variants. The glands have a different appearance and size compared to nuclei, see the images on the right, explaining this observation. In both cases, the specialist was trained based on the default SAM, which provided the best results among the non-specific models. Another architecture based on SAM has been proposed for gland segmentation (Zhang et al., 2024). However, this approach addresses semantic segmentation, not instance segmentation, so we do not compare to it here. Overall, these experiments show that neither of the available SAM variants are sufficient to address general instance segmentation tasks beyond nuclei. Finetuning for a specific task using our recipe can however be used to obtain a good tailored model.

We also evaluate PathoSAM for semantic nucleus segmentation on PanNuke (Gamper et al., 2019). For this application, we train an additional decoder based on the PathoSAM generalist, see Sec. 2.4 for details. The comparison to other methods trained for this task on PanNuke are shown in Fig. 3 c). Here, we observe that PathoSAM is the second best model, after CellViT, the clear state-of-the-art. HoverNet and HoverNext are on par and better than BioMedParse. Additional results for semantic segmentation are given in D.1 and App. D.1, reporting the individual class scores and, for PathoSAM, the result for different training strategies. Here, the main observation is that only CellViT correctly segments the "dead cells" minority class (<2 % of the nuclei). Furthermore, finetuning the encoder and training the new decoder from scratch yields the best results for PathoSAM.

### 3.4. Practical Use of PathoSAM

We evaluate the practical utility of PathoSAM by conducting a qualitative user study for interactive data annotation. Here, we check integration with two user-friendly tools: QuPath (Bankhead et al., 2017), using the SAM-API plugin (Sugawara, 2023), and $\mu$SAM (Archit et al., 2025a). We use the tools to annotate nuclei in (Janowczyk and Madabhushi, 2016), which contains nuclei in H&E stained breast tissue images. The qualitative results, comparing SAM and PathoSAM, are shown in Fig. 15 and 16. We see clear improvements with PathoSAM in both tools: it captures nuclei better from single point or box prompts.

We also implement tile-and-stitch based prediction to apply PathoSAM to whole-slide images (WSI) and enable automatic segmentation in practice. By default, we use a tile shape of $512 \times 512$ pixels, which corresponds to the training patch shape of PathoSAM, with an overlap of 64 pixels on each side. See Fig. 4 for an example and for an overview of the resources needed for WSI inference. Segmentation of a WSI runs in less than 1 hour on a GPU, requiring ca. 5 GB of VRAM per tile. It takes much longer on a CPU (ca. 37 hrs).

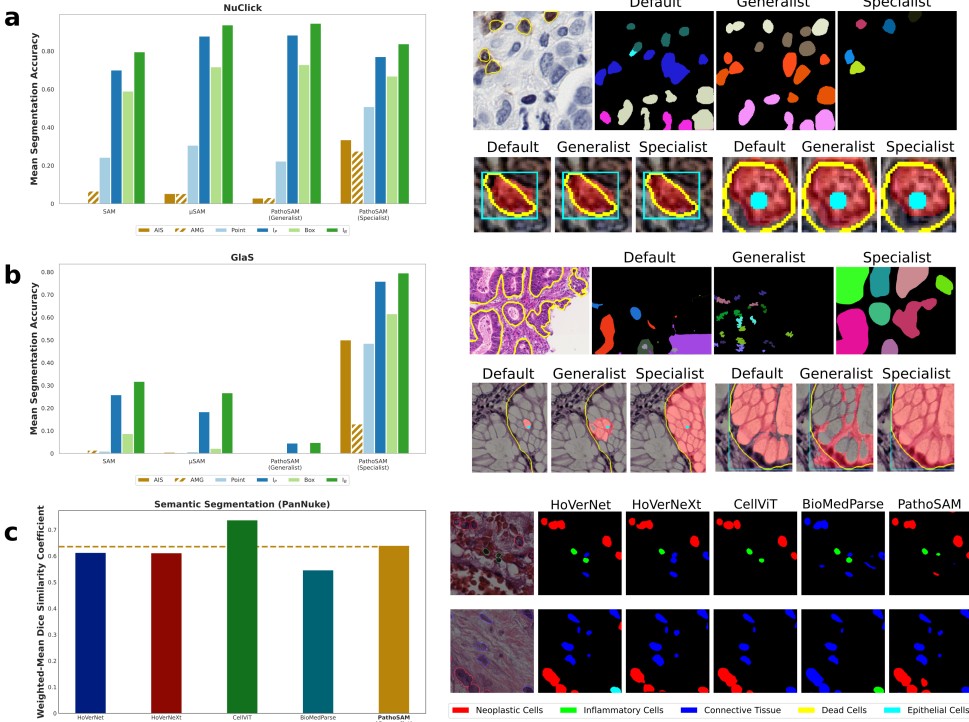

Figure 3: Segmentation of lymphocytes a) and glands b) with different SAM variants. We report measures for automatic and interactive segmentation, the right hand side shows qualitative examples from the dataset with automatic segmentation results. In both cases, PathoSAM (specialist) is trained on a separate split, using default SAM for weight initialization. c) Semantic segmentation results on PanNuke, comparing PathoSAM with other models. The right hand side shows qualitative examples from all methods.

Our implementation on the GPU has been optimized. The encoder and decoder are applied with a batch size of 1 on CPU resource, otherwise it is automatically selected depending on available VRAM. Increasing the batch size to process multiple tiles in parallel speeds up inference, especially on GPUs with large VRAM.

## 4. Discussion

Our model, PathoSAM, is the new state-of-the-art for interactive and automatic instance segmentation of nuclei in histopathology. To our knowledge, it is the first foundation model for this task, as it generalizes to diverse settings with a single model. We believe that it will have great practical impact by speeding up many nucleus analysis tasks. To support this use, we provide integration with user-friendly tools and scripts for processing WSIs.

**Limitations:** PathoSAM does not improve segmentation for other tasks in histopathology, e.g. gland or lymphocyte segmentation (see Sec. 3.3). It is thus not yet a general purpose foundation model for histopathology segmentation, but only for nucleus segmentation. However, we show that our finetuning methodology can be used to obtain improved models for other tasks as well; our tool integration provides fast data annotation for such cases. For semantic segmentation we found that PathoSAM yields good results, but that it is not yet competitive with CellViT, the state-of-the-art. We believe that this is due to the sophisticated procedure CellViT uses to oversample patches containing minority class annotations. In contrast, we train without any over-sampling or loss balancing. Notably, we have only evaluated this task on a single dataset, so it is unclear how any of the methods for semantic segmentation generalize.

**Future Work:** We plan to improve PathoSAM for semantic segmentation by (i) implementing a joint training procedure for instance and semantic segmentation, (ii) over-sampling minority classes (see prev. paragraph), and (iii) assembling a large dataset for semantic training, including annotations from (Schuiveling et al., 2025) and (Graham et al., 2019). This would provide a versatile foundation model for both instance and semantic segmentation. Furthermore, we will explore training foundation models based on SAM2 (Ravi et al., 2025). Ultimately, we would like to provide a foundation model that addresses further segmentation tasks, such as gland or tissue level segmentation, within a single model. This would however require a multi-scale segmentation approach to exploit tissue-level features that are more prominent after "zooming out".

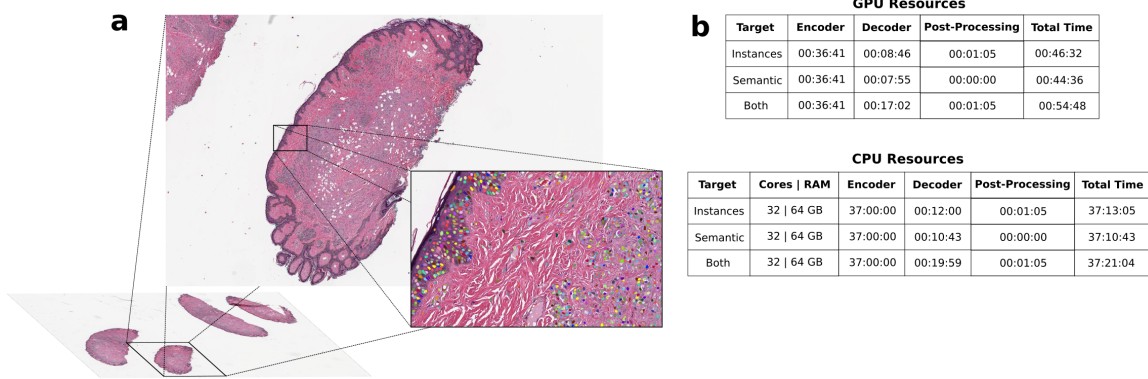

**GPU Resources**

| Target | Encoder | Decoder | Post-Processing | Total Time |
|---|---|---|---|---|
| Instances | 00:36:41 | 00:08:46 | 00:01:05 | 00:46:32 |
| Semantic | 00:36:41 | 00:07:55 | 00:00:00 | 00:44:36 |
| Both | 00:36:41 | 00:17:02 | 00:01:05 | 00:54:48 |

**CPU Resources**

| Target | Cores \| RAM | Encoder | Decoder | Post-Processing | Total Time |
|---|---|---|---|---|---|
| Instances | 32 \| 64 GB | 37:00:00 | 00:12:00 | 00:01:05 | 37:13:05 |
| Semantic | 32 \| 64 GB | 37:00:00 | 00:10:43 | 00:00:00 | 37:10:43 |
| Both | 32 \| 64 GB | 37:00:00 | 00:19:59 | 00:01:05 | 37:21:04 |

Figure 4: Segmentation on WSI. The tile shape is $512 \times 512$, with an overlap of 64 pixels. a) shows an example for automatic instance segmentation on data from https://openslide.cs.cmu.edu/download/openslide-testdata/ ($32{,}914 \times 46{,}000$ pixels). b) gives resource demand and runtime for the different inference steps using GPU (tested with an A100 with 80GB VRAM) and CPU. We indicate the resources available for the CPU; whereas the entire GPU VRAM available is utilized using batch size of 30. The minimal GPU VRAM requirement is 5GB.

## Acknowledgments

The work of Titus Griebel was supported by the Göttingen Promotionskolleg für Medizinstudierende, funded by Else Kröner-Fresenius-Stiftung. The work of Anwai Archit was funded by the Deutsche Forschungsgemeinschaft (DFG, German Research Foundation) - PA 4341/2-1. Constantin Pape is supported by the German Research Foundation (Deutsche Forschungsgemeinschaft, DFG) under Germany's Excellence Strategy - EXC 2067/1-390729940. This work is supported by the Ministry of Science and Culture of Lower Saxony through funds from the program zukunft.niedersachsen of the Volkswagen Foundation for the 'CAIMed – Lower Saxony Center for Artificial Intelligence and Causal Methods in Medicine' project (grant no. ZN4257). It was also supported by the Google Research Scholarship "Vision Foundation Models for Bioimage Segmentation". We gratefully acknowledge the computing time granted by the Resource Allocation Board and provided on the supercomputer Emmy at NHR@Göttingen as part of the NHR infrastructure, under the project nim00007. We would like to thank Sebastian von Haaren for suggestions on improving data visualizations.

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

## Appendix A. Evaluation Metric

We use the mean segmentation accuracy, following (Caicedo et al., 2019), to evaluate instance segmentation results. It is based on true positives ($TP$), false negatives ($FN$), and false positives ($FP$), which are derived from the intersection over union (IoU) of predicted and true objects. Specifically, $TP(t)$ is defined as the number of matches between predicted and true objects with an IoU above the threshold $t$, $FP(t)$ correspond to the number of predicted objects minus $TP(t)$, and $FN(t)$ to the number of true objects minus $TP(t)$. The mean segmentation accuracy is computed over multiple thresholds:

$$\text{Mean Segmentation Accuracy} = \frac{1}{|\# \text{ thresholds}|} \sum_t \frac{TP(t)}{TP(t) + FP(t) + FN(t)}.$$

Here, we use thresholds $t \in [0.5, 0.55, 0.6, 0.65, 0.7, 0.75, 0.8, 0.85, 0.9, 0.95]$. For each dataset, we report the average mean segmentation accuracy over images in the test set.

We use the dice coefficient to evaluate the semantic segmentation results. It is defined as:

$$\text{Dice Coefficient} = \frac{2 * \sum p_i \, t_i}{\sum p_i + \sum t_i},$$

for per-pixel prediction values $p_i$ and per pixel target values $t_i$. When reporting overall results for semantic segmentation, we compute the dice coefficient per class and then take a weighted sum of coefficients, with the weight corresponding to the class frequency.

Note that prior work regularly reports the panoptic quality (PQ) as metric for semantic instance segmentation on PanNuke, the dataset we use to evaluate the semantic segmentation. However, we reviewed the referenced code at https://github.com/TissueImageAnalytics/PanNuke-metrics/blob/master/utils.py and found that it is not a proper implementation of the PQ, and that it seems to only evaluate the semantic segmentation. Hence, we decided to use the Dice coefficient, which is a more established metric for semantic segmentation. We will investigate this further in future work, and if possible will introduce a true PQ metric.

## Appendix B. Dataset Description

For evaluating PathoSAM, we collected a total of 16 datasets, detailed in 1. It comprises of datasets with different tissue samples, magnifications, stainings and dimensions. We reserve the Janowczyk dataset (Janowczyk and Madabhushi, 2016) for the user study, details provided in App. D.3. Besides that, we use all datasets for either training our generalist models, evaluating them or for further downstream tasks.

## Appendix C. Architecture and Training Details

The architecture of our generalist models is shown in Fig. 5 a). It uses the original SAM architecture with the additional instance segmentation decoder, following the design of $\mu$SAM, see also Sec. 2.2.

The architecture for semantic segmentation based on the generalist is shown in Fig. 5 b). For our experiments, we use the image encoder initialized with pretrained PathoSAM model weights; the additional convolutional decoder predicts semantic class probabilities in independent channels.

We train the generalist model with the objective from $\mu$SAM (Archit et al., 2025a) that combines training of interactive and automatic segmentation (see Sec. 2.1 and Sec. 2.2). The generalist models are trained for 250,000 iterations, the PanNuke model and the specialist models for downstream tasks are trained for 100,00 iterations. We use 7 correction iterations in the training objective and a probability for using a mask prompt of 50% per iteration. This choice enables interactive both with and without addition of the mask prompt. We use a batch size of 2 and train with 40 (ViT-B), 30 (ViT-L), or 25 (ViT-H) objects per image in the batch as hyperparameters for the training objective. We use the AdamW optimizer with an initial learning rate of 1e-5. We use a learning rate scheduler that scales it by a

| Dataset | Tissue (Staining) | Num. Samples \| Magnification (Avg. Dimension) |
|---------|-------------------|-----------------------------------------------|
| **CoNSeP** | Colon (H&E) | 41 \| 40x (1000, 1000, 3) |
| *CPM15* | Brain (H&E) | 15 \| 40x, 20x (584, 668, 3) |
| *CPM17* | Brain (H&E) | 64 \| 40x, 20x (500, 500, 3) |
| **CryoNuSeg** | 10 distinct tissues (H&E) | 30 \| 40x (512, 512, 3) |
| GlaS | Colon (H&E) | 165 \| 40x (514, 760, 3) |
| Janowczyk | Breast (H&E) | 143 \| 40x (2000, 2000, 3) |
| *Lizard* | Colon (H&E) | 238 \| 20x (934, 1055, 3) |
| **LyNSec** | Lymphoma (H&E, IHC) | 600 \| 40x (512, 512, 3) |
| MoNuSAC | 4 distinct tissues (H&E) | 310 \| 40x (559, 602, 3) |
| *MoNuSeg* | 7 distinct tissues (H&E) | 51 \| 40x (1000, 1000, 3) |
| NuClick | Lymphocytes (IHC) | 871 \| 40x (256, 256, 3) |
| **NuInsSeg** | 31 distinct tissues (H&E) | 655 \| 40x (512, 512, 3) |
| *PanNuke* | 19 distinct tissues (H&E) | 7801 \| 40x (256, 256, 3) |
| *PUMA* | Melanoma (H&E) | 206 \| 40x (1024, 1024, 3) |
| **IHC TMA** | Lung (IHC) | 266 \| 40x (256, 256, 3) |
| **TNBC** | Breast (H&E) | 50 \| 40x (512, 512, 3) |

Table 1: Description of the different datasets used in our study for training generalist models (marked in *italics*), evaluating (marked in **bold**) and comparing them to other methods, and evaluating downstream tasks (marked in underline).

factor of 0.9 when the validation metric plateuas for 10 epochs. All models are trained on a single NVIDIA A100 GPU with 80GB VRAM.

## Appendix D. Segmentation Quality

### D.1. Automatic Segmentation

We provide comprehensive overview of all results for automatic instance segmentation. D.1 displays results for all 8 datasets discussed in the main paper, and additional 5 datasets, which include CPM15 (Vu et al., 2019), CPM17 (Vu et al., 2019), CoNSeP (Graham et al., 2019), LyNSeC (IHC) (Naji et al., 2024) and TNBC (Naylor et al., 2019). Further, D.1 includes detailed quantitative results for all the datasets. We provide results for all combination of models for the benchmarked methods, including all trained PathoSAM generalist models. We observe that PathoSAM is consistently either the best or among the best models for the datasets, including for out-of-domain datasets that contain different stainings (e.g. IHC staining) or imaging conditions (e.g. CryoNuSeg). For CryoNuSeg, which contains images of tissues that were cryo-sectioned, the automatic segmentation quality of the generalist model suffers from this domain shift. They also have low scores for the auto-

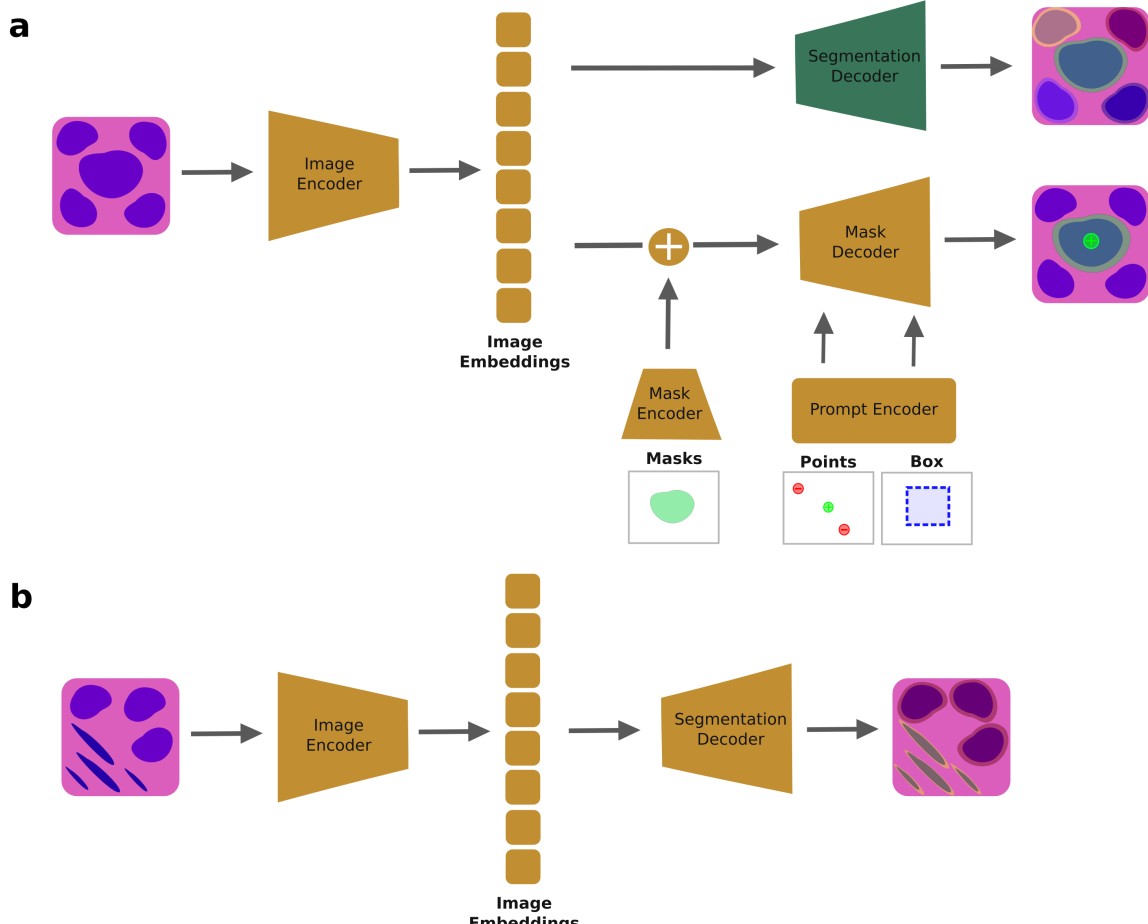

Figure 5: a) The architecture of our generalist model, which combines SAM with an additional decoder for instance segmentation, following the design from $\mu$SAM. b) The architecture for semantic segmentation, which uses a ViT-B image encoder, initialized with the weights of the PathoSAM generalist model, and an additional convolutional decoder that predicts semantic class probabilities.

matic segmentation of lymphocytes in NuClick, which is due to the fact that other nuclei are segmented as well. This dataset is used for further investigation in Sec. 3.3 to understand if training a tailored specialist model leads to an improved automatic lymphocytes segmentation.

For semantic segmentation we compare different methods that were trained for this task on the PanNuke dataset with our models finetuned based on the PathoSAM generalist (ViT-B). The results are described and discussed in Sec. 3.3. For BioMedParse, the model iteratively auto-prompts itself with modality-specific *predefined* text prompts. In this case, the model only requires the input image and the corresponding modality name (i.e. *Pathology* for our experiments). It performs an additional post-processing step to resolve potential overlaps, and combines all generated masks. Fig. 12 shows a more detailed qualitative and quantitative overview of the results.

### D.2. Interactive Segmentation

We provide a comprehensive overview of all results for interactive segmentation. Fig. 13 displays results for all 8 datasets discussed in the main paper, and 6 additional datasets, including CPM15 (Vu et al., 2019), CPM17 (Vu et al., 2019), CoNSeP (Graham et al., 2019), MoNuSaC (Verma et al., 2021), LyNSeC (IHC) (Naji et al., 2024) and TNBC (Naylor et al., 2019). Notably, MoNuSaC has additional ambiguous labels, including annotations for the cells or cytoplasm. However, we decided to include it in our evaluation to evaluate it for interactive segmentation. We clearly observe that the PathoSAM generalist outperforms SAM and $\mu$SAM for interactive segmentation with a single point or box. Considering that the model has not been trained on any other staining besides H&E, the results for IHC stained images are convincing.

Further, Fig. D.2 shows detailed quantitative results for all the above mentioned datasets. We provide quantitative results for all trained PathoSAM generalist models. Overall, we observe that our generalist model clearly outperforms SAM and $\mu$SAM, and the segmentation quality is consistent along different variants of the generalist model. For some datasets, the model trained on PanNuke (Gamper et al., 2019) performs a bit better for iterative refinement, a likely reason could be the models' distribution proximity to the training data.

### D.3. User Study

Here, we choose a H&E stained histopathology tissue image for validating SAM and PathoSAM for user-based interactive segmentation in two different tools. QuPath (Bankhead et al., 2017) expects the installation of SAM-API, which is well documented at https://github.com/ksugar/samapi. After starting the SAM-API server, the user can access the SAM extension and begin annotating. The tool currently supports default SAM (Kirillov et al., 2023) and SAM2 (Ravi et al., 2025) models. If desired, users can provide the URL for remotely stored model checkpoints, which makes it compatible with SAM-family models. Fig. 15 displays a screenshot on top for annotation with default SAM model, and bottom for annotation with PathoSAM model. We clearly see that the model identifies nuclei better with our PathoSAM generalist model, using a single point or box per object. $\mu$SAM comes with the installation package for PathoSAM, also well documented at https://computational-cell-analytics.github.io/micro-sam/ for stand-alone us-

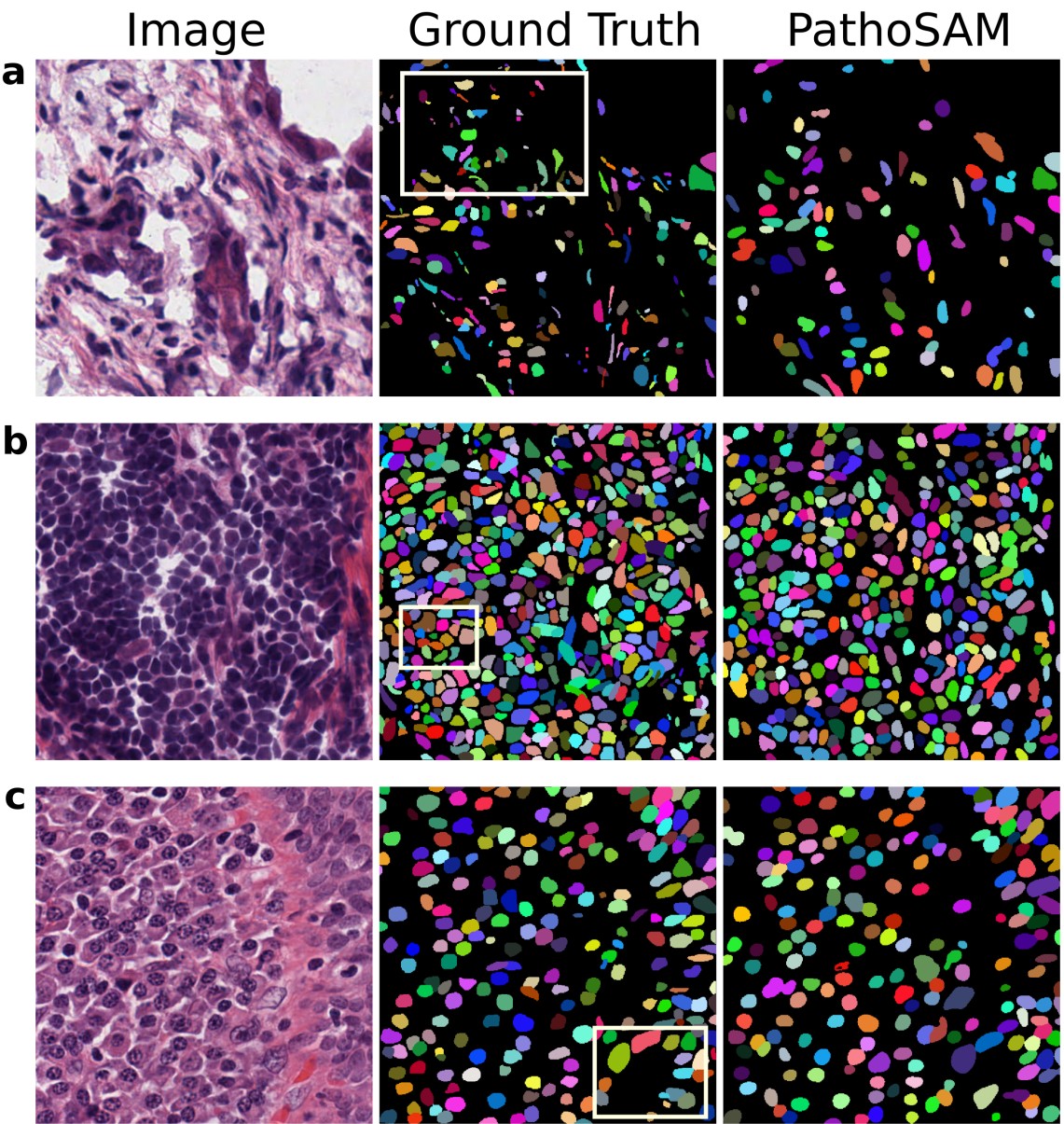

Figure 6: Three example of CryoNuSeg images for with ground-truth nucleus annotations and automatic segmentation from PathoSAM. The white box highlight the following challenges in this dataset: a) PathoSAM scores bad where the annotations are inconsistent with the nucleus ROI, b) PathoSAM struggles in regions where the foreground is hardly visible in nuclei clusters, c) PathoSAM performs well for most objects, but the scores drop due to overlapping nucleus annotations.

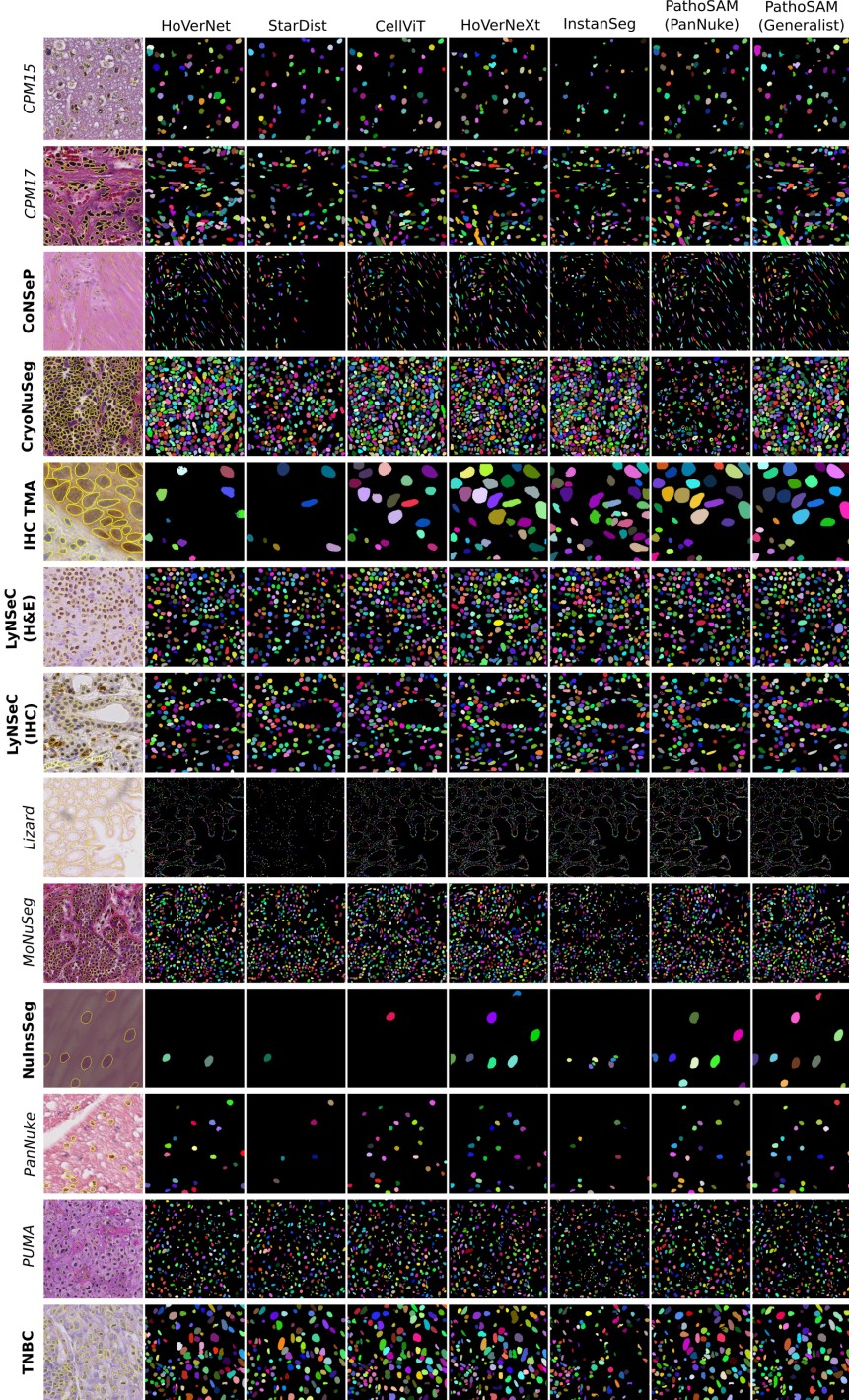

Figure 7: Qualitative plots for automatic instance segmentation with PathoSAM and other methods. Names of datasets in italics are in-domain, and bold are out-of-domain.

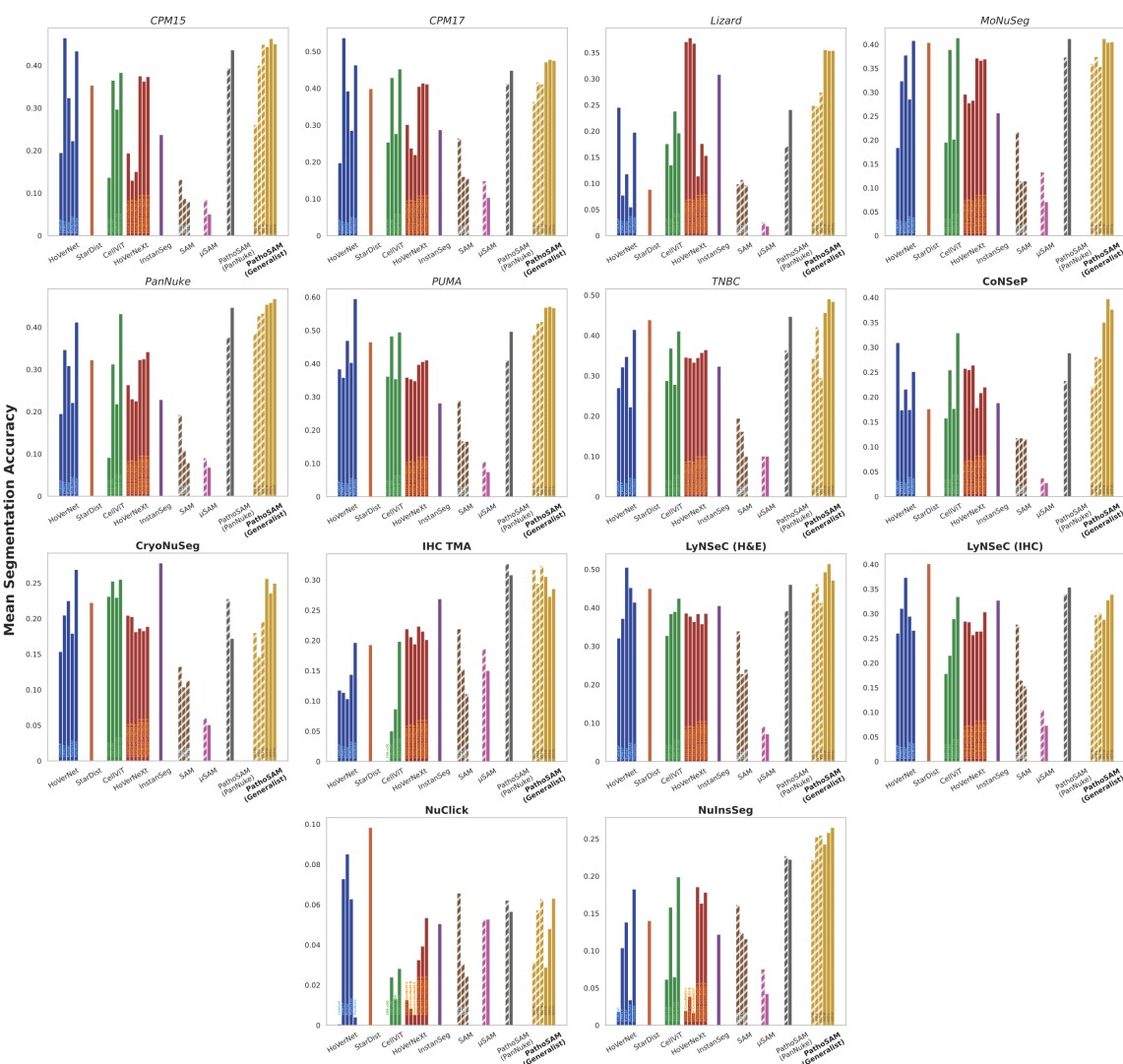

Figure 8: Quantitative results for automatic instance segmentation with PathoSAM and all other methods. Names of datasets in italics are in-domain, and bold are out-of-domain.

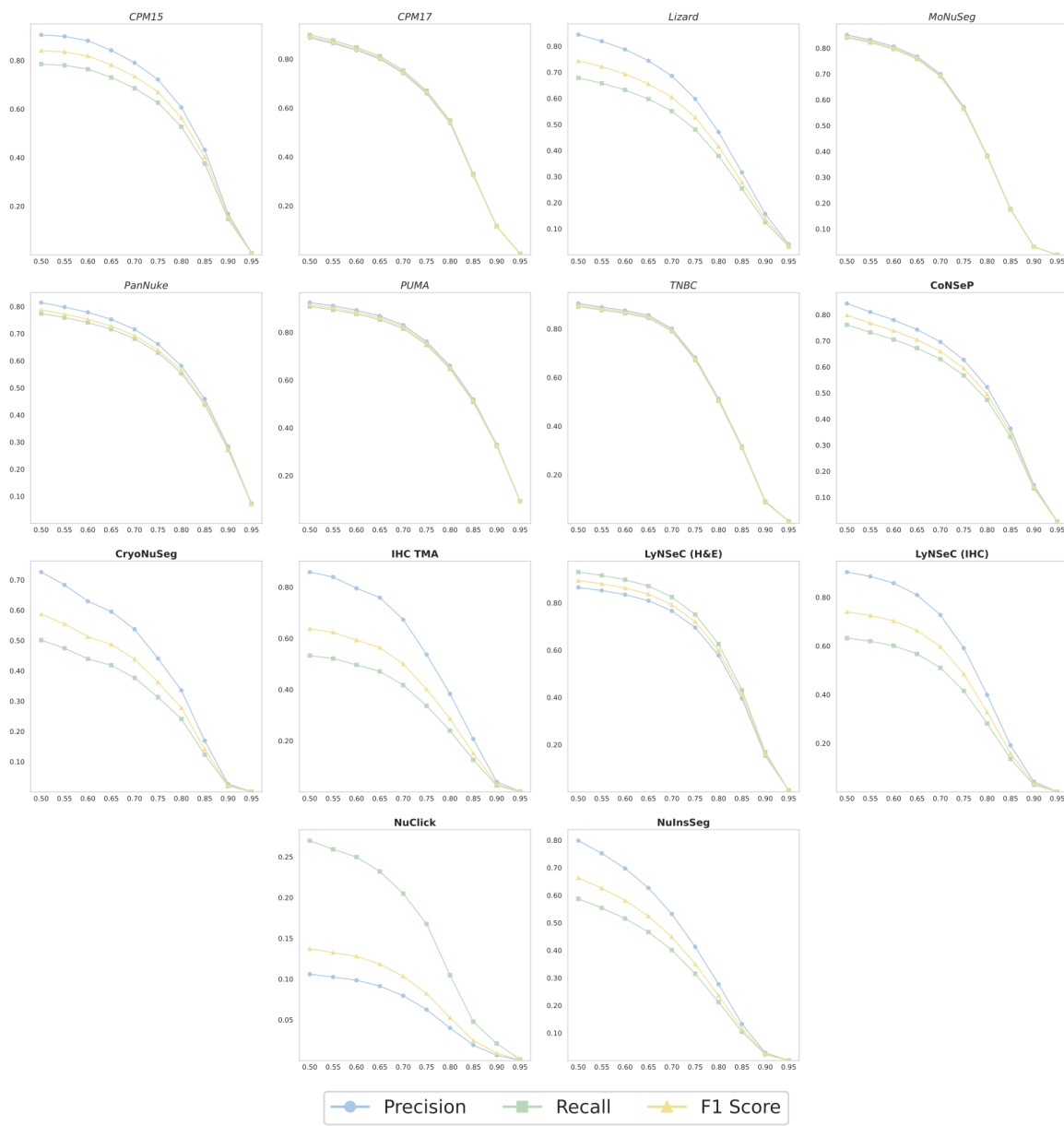

Figure 9: F1 Score, Recall and Precision for IoU thresholds from 0.5 to 0.95, for all datasets, for automatic instance segmentation with PathoSAM model (Generalist - ViT-L). Names of datasets in italics are in-domain, and bold are out-of-domain.

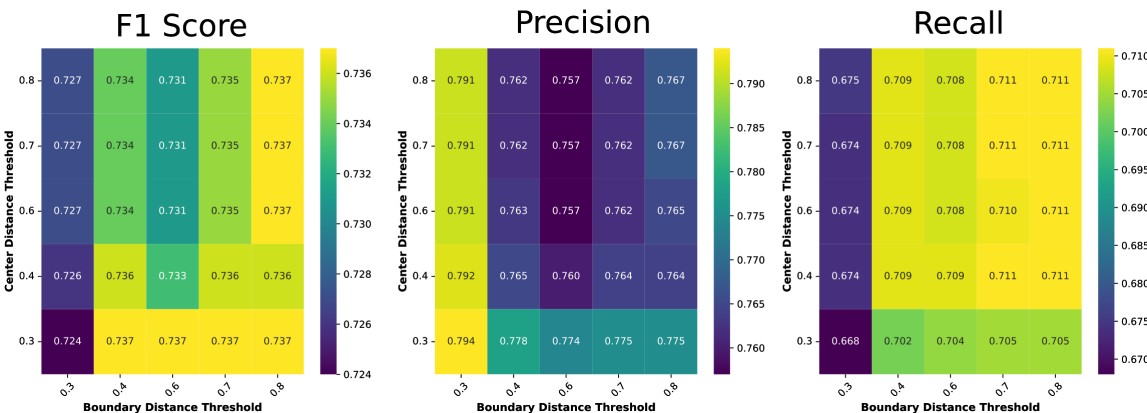

Figure 10: Grid search results over (the most important) parameters of the watershed used to obtain the instance segmentation in AIS. These parameters are the thresholds applied to center and boundary distances predicted by the segmentation decoder in order to find seeds. Please refer to (Archit et al., 2025a) for details on the instance segmentation methodology. We report the F1 Score, recall and precision for a grid-search over the two thresholds on 15 validation images of the LyNSeC (H&E) dataset, i.e. an OOD dataset. We see that precision is best for small (boundary) threshold values whereas recall is best for large thresholds. This is due to the fact that large threshold values lead to multiple seeds being derived from the distance maps, resulting in over-segmentation. Low threshold values have the opposite effect and result in under-segmentation. Hence, tuning the (boundary) threshold enables varying the degree of over-/under-segmentation.

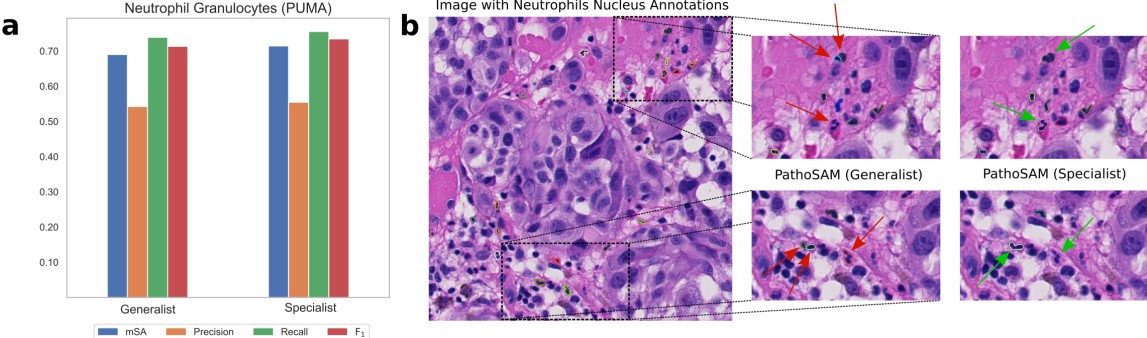

Figure 11: Qualitative and quantitative results of PathoSAM on neutrophil granulocytes, and a further specialist model trained on a few images for neutrophil granulocyte segmentation on (Schuiveling et al., 2025). Neutrophils have multi-lobed nuclei, which are more challenging to segment compared to "regular" nuclei. a) The scores are on par, with a slight favor for the specialist model. Note: we report scores specifically for neutrophil granulocyte instance segmentation, excluding other nucleus types from the evaluation. b) An example image with annotations for neutrophils. The zoom-ins show the segmentation results for the generalist model (middle) and the neutrophil specialist model (right). Arrows indicate neutrophils that are segmented better by the specialist compared to the generalist. These results indicate that both models can segment neutrophils reasonably well, and that the specialist model performs a bit better than our generalist. Importantly, this shows that our method can segment such complex shapes correctly, and that in practice the quality of results depends on the presence and quality of the respective annotations in the training data.

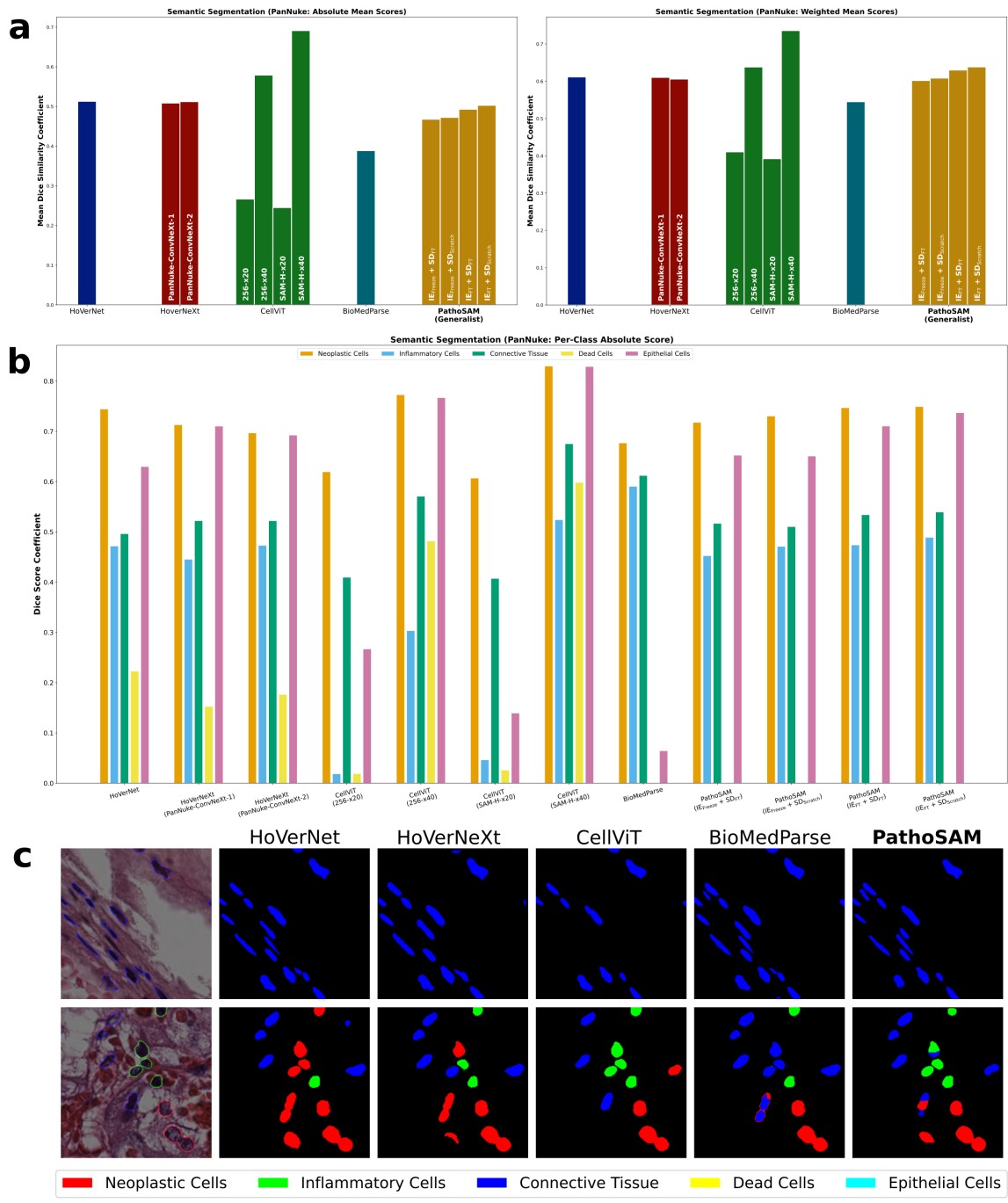

Figure 12: Quantitative results for semantic segmentation, for different versions of the benchmarked methods (a) and the per-class performance for the respective best method (b). c) shows qualitative examples.

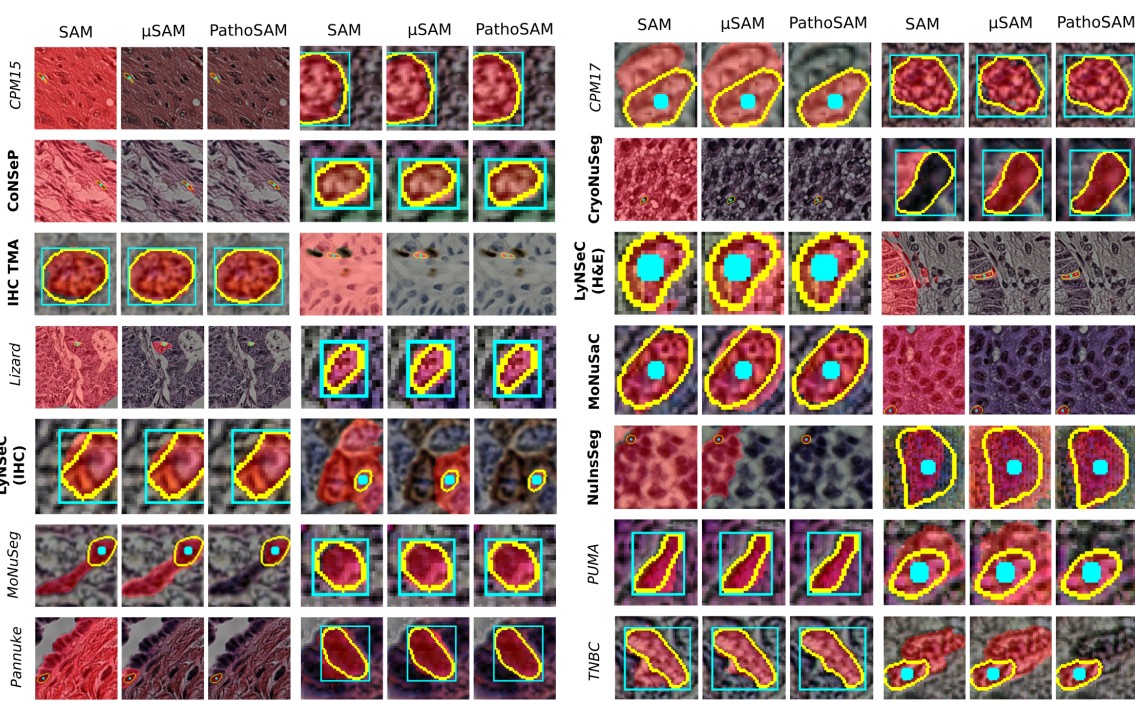

Figure 13: Qualitative plots for interactive segmentation with SAM, $\mu$SAM and PathoSAM ViT-B generalist model.

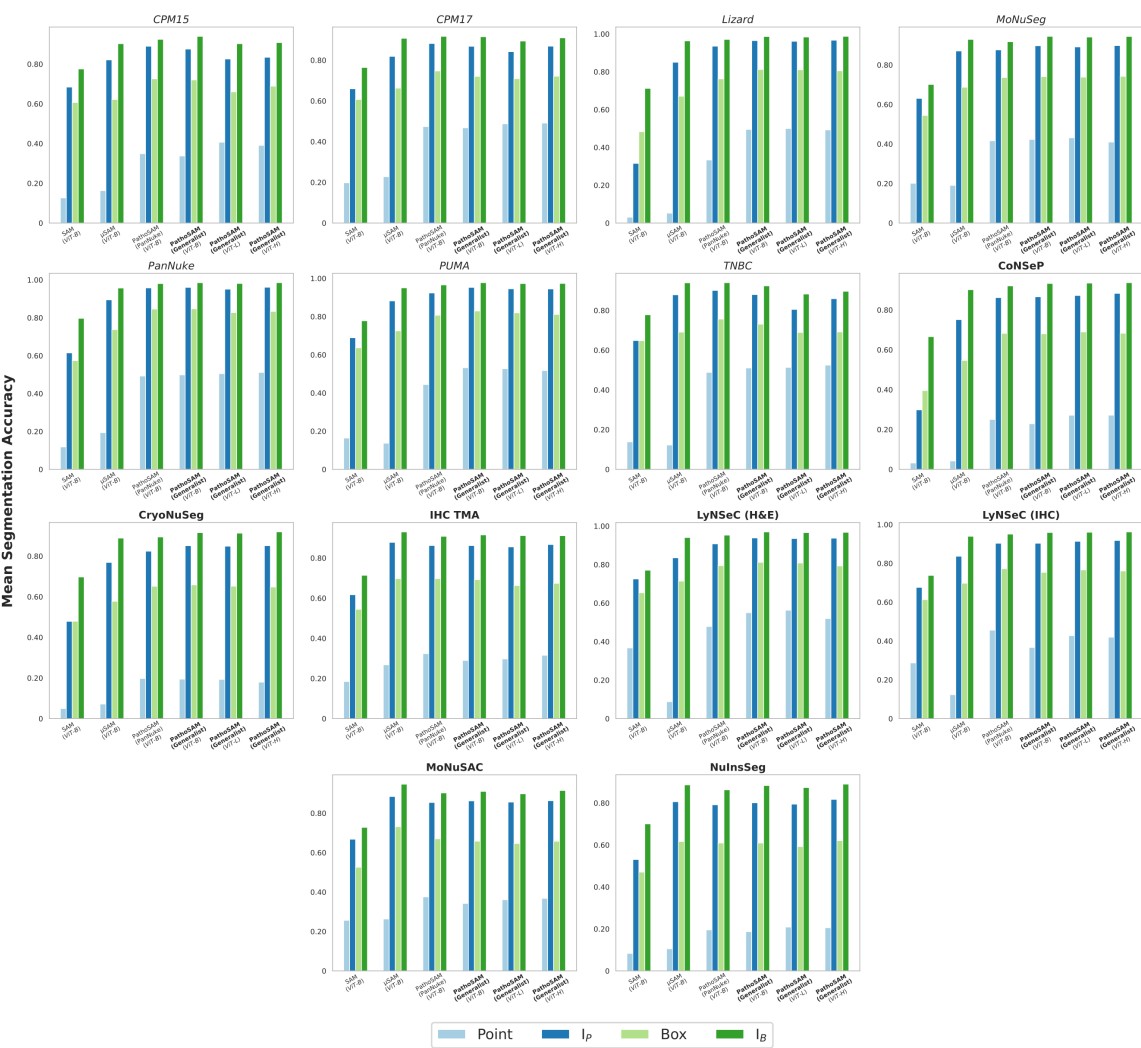

Figure 14: Quantitative results for interactive segmentation for SAM, $\mu$SAM, PathoSAM (PanNuke) and all variants of PathoSAM (Generalist) models.

age. A user has the flexibility to start napari (Sofroniew et al., 2024) and select the *Segment Anything for Microscopy* available under extensions and a desired annotator (for histopathology images, the relevant choice is "Annotator 2D"), or access the 2D annotator with CLI scripts. For images larger than the training data, which is quite common for WSIs, $\mu$SAM supports segmentation over tiles over the entire image. We make use of this feature to evaluate both SAM and PathoSAM models. In addition, PathoSAM provides a simple CLI script to run automatic segmentation over WSIs.

Overall, our observation affirms PathoSAM's interactive segmentation capability and tool compatibility across popular annotation tools. We believe PathoSAM will speed up such annotation workflows.

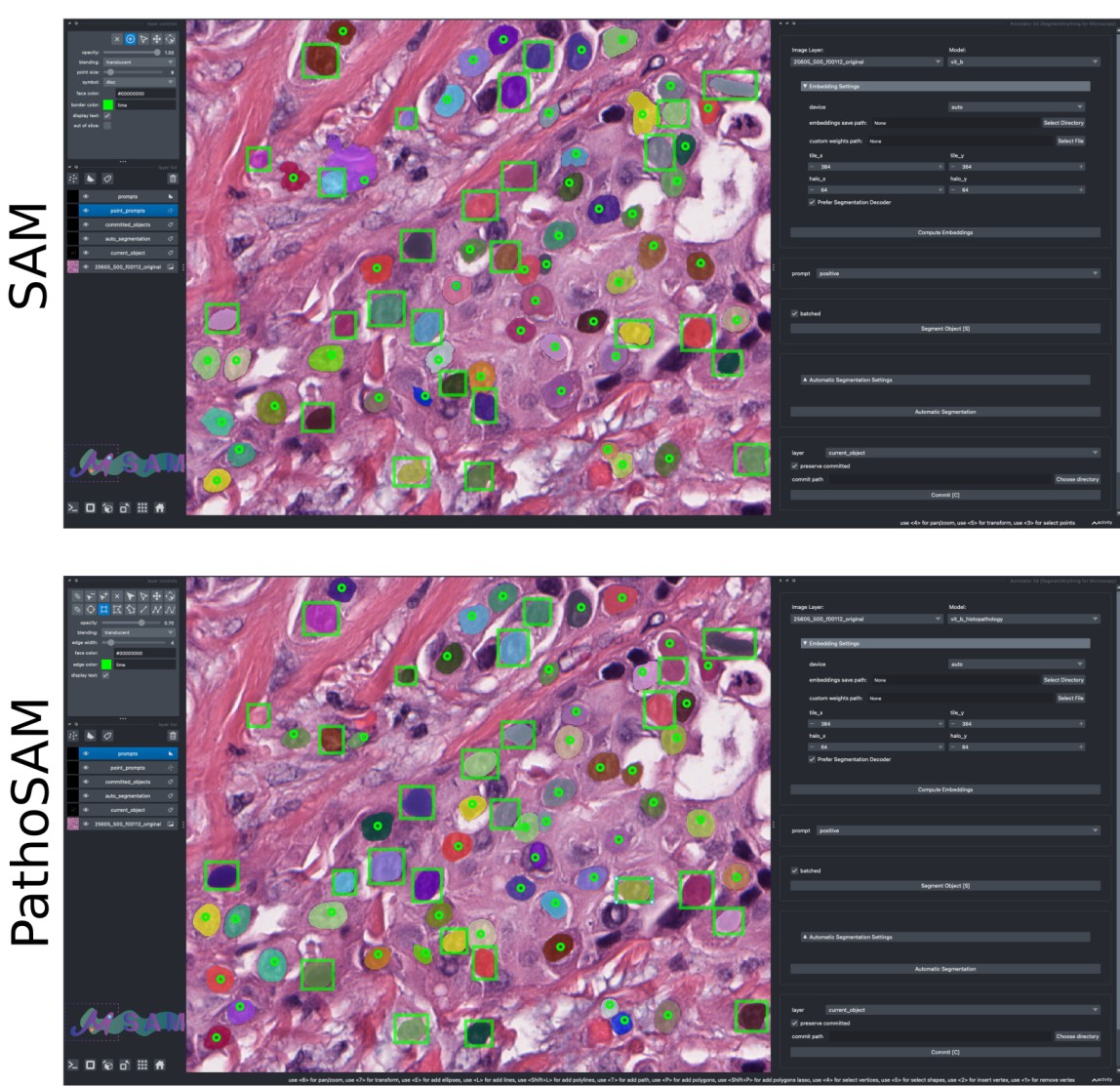

Figure 15: User study with $\mu$SAM in napari, with tiling-window based inference for this large image

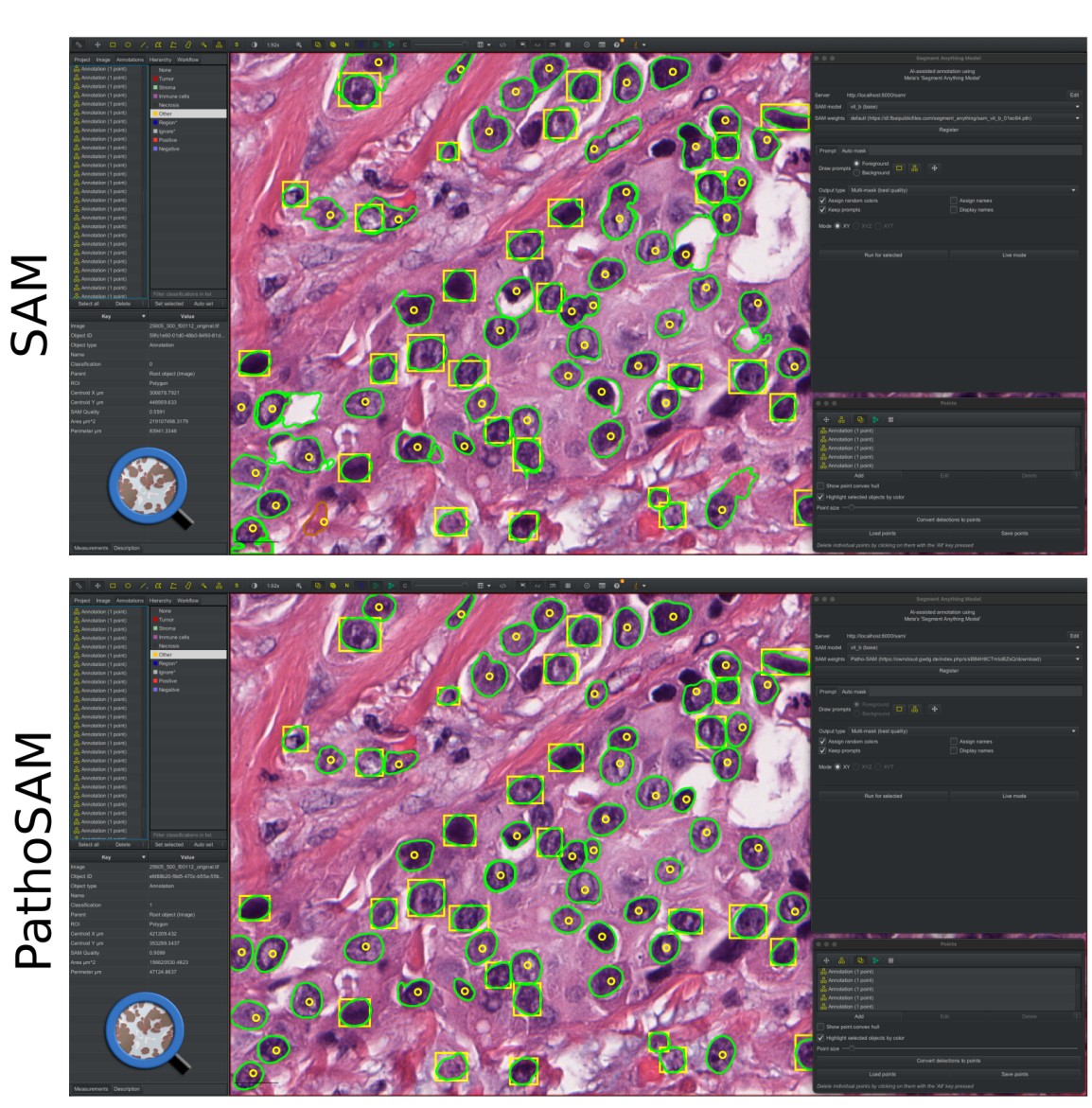

Figure 16: User study with SAM-API in QuPath

