# OpenReview forum: "Segment Anything for Histopathology"
_MIDL.io/2025/Conference — MIDL 2025 Oral_

### Official Review · Reviewer_2ibn · 2025-02-20

**Confidence:** 3
**Preliminary Rating:** 4
**Recommendation:** Poster
**Final Rating:** 5

**Summary:**

This paper introduces PathoSAM, a specialized adaptation of the Segment Anything Model (SAM) for histopathological nucleus segmentation. The authors demonstrate that by finetuning SAM on six diverse, publicly available H&E datasets, they can produce both an interactive and automatic approach for identifying nuclei, and compare it against established baselines. To broaden its usefulness, they add a separate decoder for semantic (multi-class) segmentation, showing preliminary results that, while strong, still slightly lags behind CellViT on PanNuke. They also integrate PathoSAM with user-oriented tools like QuPath and µSAM, and discuss how it can be applied to whole-slide images (WSIs). The paper makes a convincing case for a foundation-style model that can handle multiple histopathological datasets more robustly than the generic SAM, helping reduce the costs of manual annotation in high-stakes environments like digital pathology.

**Strengths:**

- By addressing nucleus segmentation in a broad set of histopathological images, the paper meets a well-known need in digital pathology where domain shifts (different stains, tissue types, scanning resolutions) often degrade performance of single-dataset models.
- Training was done on six distinct H&E datasets, then PathoSAM was tested on eight additional out-of-domain datasets, including IHC-stained tissue, cryo-sectioned tissue, and varied morphological targets. This underscores generalizability.
- By combining the original SAM framework (interactive prompts) with a µSAM-inspired decoder for automatic instance segmentation, the paper unifies two important workflows (manual refinement and batch processing). This dual capability is valuable in pathology, where both routine bulk segmentation and careful human-in-the-loop correction are needed.
- The paper systematically compares PathoSAM to specialized nucleus-segmentation methods using instance-level metrics and standard thresholds. Such quantitative evaluation is very helpful for verifying new claims in the bio-imaging domain.
- Integration with QuPath and µSAM could encourage adoption by pathologists and domain experts.

**Weaknesses:**

- While the paper’s semantic segmentation extension is promising, it does not consistently outperform state-of-the-art methods like CellViT on challenging classes (e.g., dead cells). This could indicate some limitations in handling minority classes, which are important in many pathology applications.
- In some domain-shifted settings, particularly cryo-sections or highly variable IHC images, PathoSAM seems to either omit subtle nuclei or include excess background objects. Although the authors highlight these cases qualitatively, a more thorough quantitative breakdown of false positives vs. false negatives would be interesting.
- The authors mention GPU usage (e.g., A100 with 80 GB) but do not provide information about the memory constraints for WSI inference in real clinical pipelines. Some guidelines or performance analyses for tiling, overlap, and speed considerations could be valuable for applying the proposed approach.
- A discussion on the limitations as well as future directions is missing.

**Detailed Comments:**

- Providing an explicit table of performance trade-offs under different prompt-sampling schemes (e.g., ratio of point vs. box prompts during training) would clarify how best to replicate or further tune interactive segmentation.
- The user study references in QuPath and µSAM are a strong addition, but it might help to include actual annotation times or confusion metrics in a more formal user study protocol (e.g., average corrections required, time per annotation).
- While the authors highlight how PathoSAM can over-segment in some contexts, a small set of best practices for addressing over-segmentation might be valuable in bridging the gap between good research results and robust real-world application.
- The huge amount of very interesting results in different evaluation settings is on the one hand impressive and on the other hand making it hard to grasp the main message of your work. I'd prefer to have the result comparison given in tables instead of bar plots (which are quite small). But of course this is a subjective view.
- If you are looking for further data sets to challenge PathoSAM, you could have a look at this: https://doi.org/10.1038/s41597-020-00608-w

**Justification Of The Final Rating:**

Thank you very much for taking the time to answer the questions raised! I really appreciate the effort that has been made to resolve all my concerns. I am therefore changing my rating to strong accept, which is also in line with my fellow reviewers. Looking forward to see the future path of you work.

**Justification Of The Preliminary Rating:**

This paper presents a valuable adaptation of SAM for histopathology, offering both automatic and interactive segmentation modes and showing strong performance across diverse datasets. There are some open questions, but I nonetheless see it as a valuable contribution regarding histopathologic image segmentation.

**Questions To Address In The Rebuttal:**

- What are the typical memory footprints and inference times especially for whole-slide images under different tiling strategies in practice? Are there recommended overlaps or tile sizes to balance speed and consistent object IDs across tile boundaries?
- Do you have any plans or preliminary results for training a single universal decoder that performs both instance and semantic segmentation simultaneously, similar to existing multi-task histology frameworks? If so, what challenges do you foresee?
- Please describe the limitations of your current approach and give us a hint on your future plans.

---

> ### Author Response · Authors · 2025-03-08
>
> Thank you for the detailed review of our work, the correct and overall positive assessment, and the good feedback on how to further improve it. Regarding the comments / weaknesses:
>
> > R3: While the paper’s semantic segmentation extension is promising, it does not consistently outperform state-of-the-art methods like CellViT on challenging classes (e.g., dead cells). This could indicate some limitations in handling minority classes, which are important in many pathology applications.
>
> Indeed, our model does not perform as good as CellViT for infrequent classes like dead cells. We strongly believe that this is due to the sophisticated oversampling procedure that CellViT uses to over-represent patches with minority classes; see also the response to R1. Note that we did not have time to test this hypothesis, but will implement the oversampling logic in the future, among other planned improvements for semantic segmentation.
>
> > R3: In some domain-shifted settings, particularly cryo-sections or highly variable IHC images, PathoSAM seems to either omit subtle nuclei or include excess background objects. Although the authors highlight these cases qualitatively, a more thorough quantitative breakdown of false positives vs. false negatives would be interesting.
>
> We have added Figure 9 to the appendix to provide further details on these points. It shows the evolution of precision, recall and F1-score as a function of the overlap threshold used to compute false positives, false negatives and true positives, see the appendix section on evaluation metrics for details. This figure gives a more detailed overview of the segmentation quality for all datasets.
>
> > R3: The authors mention GPU usage (e.g., A100 with 80 GB) but do not provide information about the memory constraints for WSI inference in real clinical pipelines. Some guidelines or performance analyses for tiling, overlap, and speed considerations could be valuable for applying the proposed approach.
>
> We have added a new section to the main text that discusses practical usage of our model, both within user-friendly tools and for automatic processing of WSIs. This includes the new Figure 4, which shows an example image for application to WSIs and gives an overview of the runtimes on GPU and CPU. Note that our GPU implementation is not yet optimized, as we could stack tiles across the batch axis to profit from GPUs with large VRAM. We will implement this improvement for the camera-ready version and update the runtime analysis accordingly.
>
>
> > R3: A discussion on the limitations as well as future directions is missing.
>
> Thank you for pointing this out. We have added a paragraph on future research directions and limitations in the discussion section

---

> > ### Author Response · Authors · 2025-03-08
> >
> > Regarding the detailed comments:
> >
> > > R3: Providing an explicit table of performance trade-offs under different prompt-sampling schemes (e.g., ratio of point vs. box prompts during training) would clarify how best to replicate or further tune interactive segmentation.
> >
> > This is a valuable suggestion. However, due to the time constraints of the revision and the long training times of the foundation models (ca. 1 week on an A100 GPU), we cannot address this now. We have performed a similar analysis, comparing training with different prompt-sampling strategies for medical imaging (excluding histopathology) in other recent work from the lab: https://arxiv.org/abs/2501.11734.
> > We would assume that our findings, a complex training scheme that mimics interactive annotations being critical for good interactive segmentation, hold for histopathology data as well.
> >
> > > R3: The user study references in QuPath and µSAM are a strong addition, but it might help to include actual annotation times or confusion metrics in a more formal user study protocol (e.g., average corrections required, time per annotation).
> >
> > Thank you for the suggestion. We indeed believe that such a study would be very helpful, but it is out of scope for our current contribution. In general, such a study would be more suitable for a journal application that studies the practical applications of segmentation / foundation models in more depth.
> >
> > > R3: While the authors highlight how PathoSAM can over-segment in some contexts, a small set of best practices for addressing over-segmentation might be valuable in bridging the gap between good research results and robust real-world application.
> >
> > We have added Figure 10 to the appendix, which shows precision vs. recall, corresponding to under- vs over- segmentation as a function of the two most important parameters for instance segmentation post-processing.
> >
> > > R3: The huge amount of very interesting results in different evaluation settings is on the one hand impressive and on the other hand making it hard to grasp the main message of your work. I'd prefer to have the result comparison given in tables instead of bar plots (which are quite small). But of course this is a subjective view.
> >
> > Thank you for pointing this out. Creating easy-to-understand tables with an overview of all results is also quite challenging due to the many different experiments (several datasets and several models per dataset, including the same model architecture trained on different datasets for some models). Hence we have kept the current bar plots. However, we will increase the font size in the plots for the camera ready version in order to improve their readability.
> >
> > > R3: If you are looking for further data sets to challenge PathoSAM, you could have a look at this: https://doi.org/10.1038/s41597-020-00608-w
> >
> > Thank you for suggesting this dataset. However, it contains nuclei imaged in fluorescence microscopy, and not histopathology. This is out of scope for the PathoSAM models. Please note that the light microscopy generalist model from microSAM (our prior work) already addresses the segmentation of nuclei in (immuno)fluorescence data. See https://www.nature.com/articles/s41592-024-02580-4 for details.
> >
> > Regarding the questions to address in the rebuttal:
> >
> > > R3: What are the typical memory footprints and inference times especially for whole-slide images under different tiling strategies in practice? Are there recommended overlaps or tile sizes to balance speed and consistent object IDs across tile boundaries?
> >
> > We have addressed this in Figure 4, see above.
> >
> > > R3: Do you have any plans or preliminary results for training a single universal decoder that performs both instance and semantic segmentation simultaneously, similar to existing multi-task histology frameworks? If so, what challenges do you foresee?
> >
> > We indeed plan to train a joint model for instance and semantic segmentation. Note that this model would still have two separate decoders, one for instances, one for semantics, i.e. using the same architecture as we employ now. However, we plan to train it within a single training framework that jointly optimizes interactive, instance, and semantic segmentation. Currently, we train in two separate steps, first for interactive and instance segmentation, then for semantic segmentation. Joint training will likely result in a model with a more balanced performance across all three tasks. The main challenge here is the fact that only a subset of the available data has both semantic and instance segmentations. Hence, we will need to add a flexible training framework to accommodate these data limitations.
> >
> > > R3: Please describe the limitations of your current approach and give us a hint on your future plans.
> >
> > We have added a paragraph on this in the discussion. Note that the joint model (previous answer) is part of the future plans.

---

> > > ### Comment · Reviewer_2ibn · 2025-03-12
> > >
> > > Thank you very much for taking the time to answer the questions raised! I really appreciate the effort that has been made to resolve all my concerns. I am therefore changing my rating to strong accept, which is also in line with my fellow reviewers. Looking forward to see the future path of you work.

---

### Official Review · Reviewer_xzPo · 2025-02-21

**Confidence:** 4
**Preliminary Rating:** 5
**Recommendation:** Oral

**Summary:**

This paper proposes Segment Anything for Histopathology (PathoSAM), which builds upon the original SAM framework and extends to applications such as semantic segmentation and nucleus segmentation. Extensive experimental results show that PathoSAM is competitive in performance for automatic instance segmentation, as well as interactive segmentation.

**Strengths:**

The article overall is well written and ideas have been clearly illustrated. Extensive results have covered use cases for both automatic segmentation and interactive segmentation. Training and benchmarking datasets cover a wide ranger of tissue types and fixation types. The proposed tool can have good impact because of its integration with routine pathology software such as QuPath.

**Weaknesses:**

- Minor thing: it would be nice to not discontinue the paragraph in D.3. on page 17 until page 24.
- Would be interesting to give examples of neutrophil nuclei, since they are multi-lobed and have a unique nuclear envelope composition.

**Detailed Comments:**

N/A

**Justification Of The Preliminary Rating:**

The article overall is well written and ideas have been clearly illustrated. Extensive results have covered use cases for both automatic segmentation and interactive segmentation. Training and benchmarking datasets cover a wide ranger of tissue types and fixation types. The proposed tool can have good impact because of its integration with routine pathology software such as QuPath.

**Questions To Address In The Rebuttal:**

- Would like to see if the model makes a split error for neutrophil nuclei.

---

> ### Author Response · Authors · 2025-03-08
>
> Thank you for the correct summary and positive assessment of our work. Regarding the weaknesses / comments:
>
> > R2: Minor thing: it would be nice to not discontinue the paragraph in D.3. on page 17 until page 24.
>
> Thank you for pointing this out. We have fixed the formatting of the appendix to amend the issue.
>
> > R2: Would be interesting to give examples of neutrophil nuclei, since they are multi-lobed and have a unique nuclear envelope composition.
>
> Thank you for this suggestion. We have now investigated segmentation of neutrophil nuclei with our model and added Figure 11 to the appendix with the results. Briefly summarized, we find that our model can in principle segment these nuclei, despite their complex shape. We show that a specialist model, trained specifically for neutrophil segmentation, leads slightly better results, indicating that the practical performance on this task depends on the prevalence and quality of the respective annotations in the training data.

---

### Official Review · Reviewer_yUxc · 2025-02-22

**Confidence:** 3
**Preliminary Rating:** 5
**Recommendation:** Poster
**Final Rating:** 5

**Summary:**

This paper fine-tunes the SAM on various nucleus segmentation datasets to build a  foundation model for nucleus segmentation: PathoSAM. It supports both automatic (by providing grid points) and interactive (prompt by mask/point/bounding box) nucleus instance segmentation in histopathology.

**Strengths:**

Although the technique novelty of the proposed method is very limited, it builds a foundation model for nucleus segmentation. The authors conducted extensive experiments to show the effectiveness of the proposed method. It seems work well on a variety of datasets and tasks. The authors also mentioned that it can work with popular tools for data annotation. This is very useful for the computational pathology society.

**Weaknesses:**

1. The technique novelty of the proposed method is very limited. It fine-tunes SAM using some existing method.
2. Can the authors provide some possible explanations on why the proposed method does not beat the SOTA, CellViT?
3. The authors evaluated the performance of the proposed method on some gland segmentation tasks. How does it compared with other fine-tuned SAM on such tasks, e.g. Zhang et al., 2023 ?

**Detailed Comments:**

The fonts in most figures (e.g. Figure 1 (b) (c) and Figure 2) are generally too small.

**Justification Of The Final Rating:**

As I mentioned in the initial review, the limited novelty of this paper does not affect its wide application in the computational pathology society. The rebuttal solves most of my concerns and thus I keep my initial rating of "Strong accept".

**Justification Of The Preliminary Rating:**

As I mentioned in the strength section, although the technique novelty of the proposed method is very limited, the proposed method is very useful in practice. The authors also conducted extensive experiments to show the effectiveness of it.

**Questions To Address In The Rebuttal:**

As mentioned in the weakness section.

---

> ### Author Response · Authors · 2025-03-08
>
> Thank you for the correct summary and positive appraisal of our work. Regarding the weaknesses mentioned in your review:
>
> > R1: 1. The technique novelty of the proposed method is very limited. It fine-tunes SAM using some existing method.
>
> We agree that our contribution does not introduce any fundamental new algorithms or methodological innovation. Rather, we build on (our) prior work that has demonstrated how to finetune SAM for interactive and automatic instance segmentation in a given domain, including additional functionality for semantic segmentation. We then aggregate a very large dataset for nucleus segmentation, comprising several open-source datasets, train PathoSAM on this dataset, and demonstrate that the resulting model is the state-of-the-art for interactive and automatic instance segmentation in histopathology, using a **fair** comparison with the most important reference methods. We also evaluate semantic segmentation, where we find that our method is among the best available approaches, second only to CellViT, the state-of-the-art for semantic segmentation. Given these contributions, especially the combination of state-of-the-art interactive and automatic instance segmentation, we believe that our method will have great practical value in speeding up nucleus segmentation for new tasks / datasets. To this end we make it easy to apply in practice through user-friendly tool integration (QuPath and microSAM) and inference functionality for WSI.
>
> > R1: 2. Can the authors provide some possible explanations on why the proposed method does not beat the SOTA, CellViT?
>
> First, we would like to clarify that our method beats CellViT for automatic instance segmentation. However, it indeed scores lower for automatic semantic segmentation. We believe that the reason for this is mainly due to the training procedure of CellViT. They employ a sophisticated strategy to oversample patches that contain nuclei of minority classes (e.g. dead cells). In contrast, we did not use any such over-sampling logic. We believe that this is the main explanation for this difference. However, we did not have time to implement this sampling logic now to test this hypothesis. We plan to improve our model for semantic segmentation in the future by implementing oversampling, training on a larger dataset, and using a common framework for semantic and instance segmentation. See the new paragraph on future work for details.
>
> > R1: The authors evaluated the performance of the proposed method on some gland segmentation tasks. How does it compared with other fine-tuned SAM on such tasks, e.g. Zhang et al., 2023?
>
> Thank you for pointing out this reference, we were previously not aware of it and now cite it in the paper. Note, however, that Zhang et al. solve a different task compared to us: they perform semantic segmentation of glands, whereas we perform instance segmentation. Due to this difference, and the limited time for experiments in the rebuttal phase, we chose not to compare to it.

---

### Author Rebuttal · Authors · 2025-03-08

**Rebuttal:**

Dear Reviewers, dear Area Chair,

We thank you for your thorough reviews and the positive appraisal of our work. Based on the reviewer comments we have made the following changes:
- We have added a section on practical applications to the main text, including Figure 4 that shows an example image for WSI inference with our method and that gives the processing times of this image with GPU and CPU (addressing a request by R3).
- We have updated the discussion to clearly outline the limitations and plans for future work (addressing a request by R3).
- We have added Figure 9 to the Appendix. It shows the curves of precision, recall and F1 score over overlap thresholds of the metric, to enable a better appraisal of the segmentation quality and address a request by R3.
- We have added Figure 10 to the Appendix, which shows the dependence of precision and recall as a function of the most important parameters of instance segmentation postprocessing, also addressing a request by R3.
- We have added Figure 11 to the appendix, which investigates the segmentation quality for neutrophils, addressing a request by R2.

All changes in the text, including the captions of figures we have added, are marked in red.

**Supporting Material:**

/attachment/f4d20c05c2b19362292840b8dee868104f1b87b2.pdf

---

### Comment · Area_Chair_UMYp · 2025-03-08

Dear MIDL Reviewers, the discussion stage (March 8–14, 2025) begins today. The authors' rebuttal has been uploaded to OpenReview, and you are encouraged to engage with them for any necessary clarifications. Your participation in the discussion is greatly appreciated.

---

### Meta-Review · Area_Chair_UMYp · 2025-03-15

**Recommendation:** Accept (Oral)
**Confidence:** 5

**Metareview:**

All reviewers gave positive rating.